# Understanding the hemodynamic changes in fetuses with coarctation of the aorta using a lumped model of fetal circulation

Inmaculada Villanueva-Baxarias[1], *, Anna Pellisé-Tintoré[1], María Pérez-Rodríguez[1], Laura Nogué[2], Pooja Vaziraani[2], Iris Soveral[2, 3], Fàtima Crispi[2, 4], Olga Gómez[2, 4], Patricia Garcia-Canadilla[2, 5], Oscar Camara[1], Bart Bijnens[1, 6], Gabriel Bernardino[1]

**1** Department of Engineering, Universitat Pompeu Fabra, Barcelona, Spain, **2** BCNatal Fetal Medicine Research, Hospital Clínic de Barcelona, Barcelona, Spain, **3** Consorci Sanitari Integral, Hospital General de l'Hospitalet, Barcelona, Spain, **4** Institut d'Investigacions Biomèdiques August Pi I Sunyer, Barcelona, Spain, **5** Interdisciplinary Cardiovascular Research Group, Institut de Recerca Sant Joan de Déu, Esplugues de Llobregat, Spain, **6** ICREA - Catalan Institution for Research and Advanced Studies, Barcelona, Spain

* mariainmaculada.villanueva@upf.edu

## Abstract

Coarctation of the aorta (CoA) is a common congenital heart defect characterized by aortic narrowing. Prenatally, it has mild hemodynamic effects as right ventricular disproportion and ductus arteriosus (DA) dilation occur as adaptive mechanisms, but their impact on CoA hemodynamics remains poorly understood. To investigate this, we built a closed 0D computational model of fetal circulation and simulated different CoA cardiovascular remodeling patterns, including aortic isthmus (AoI) narrowing, ventricular disproportion, and DA dilation. Our results showed mild AoI narrowing (80% of reference diameter) required up to 1.7 right/left ventricular end-diastolic volume ratio and 115% DA dilation to maintain physiological pressures, wall shear stresses, and organ perfusion. In contrast, severe narrowing (20% of reference AoI diameter) required up to 5 right/left ventricular end-diastolic volume ratio and 125% DA dilation, highlighting the necessity of co-occurrence of prenatal ventricular disproportion and DA dilation to compensate for AoI narrowing. These physiological regions were validated with ultrasonographic measurements from 7 controls and 9 CoA patients. We compared blood pressures, velocities, and volumetric flow rates across different fetoplacental anatomical sites. AoI velocity showed a delayed retrograde flow peak and increased antegrade diastolic velocity with greater AoI narrowing, which may aid in diagnosing CoA. Minimal differences were observed in other velocities and pressures. Volumetric flow rates across varying degrees of AoI narrowing decreased in the AoI and mitral and aortic valves, remained stable in the middle cerebral and umbilical arteries, and increased in the DA and tricuspid and pulmonary valves. Therefore, we corroborated that in fetal CoA a redistribution of blood flow occurs to ensure perfusion of the brain and placenta, without a significant alteration

**Data availability statement:** All relevant data are submitted as Supporting Information, and all code used for running experiments is available on a GitHub repository at https://github.com/InmaV/CoA_scenarios.git, DOI: 10.5281/zenodo.14826951

**Funding:** This work was supported by European Social Fund Plus (ESF+) and Generalitat de Catalunya (Joan Oró Grant 00237 to I.V.B.; Beatriu de Pinos Grant 2018-BP-00201 to P.G.C.), by the H2020 Marie Skłodowska-Curie Actions (Marie Skłodowska-Curie Grant 801370 to P.G.C), by ESF+ and Ministerio de Ciencia e Innovación (RYC2023-043724-I to P.G.C. and RYC2022-035960-I to G.B.), and by Fundació la Marató de TV3 (Project CoALeS 650/C/2020). The funders had no role in study design, data collection and analysis, decision to publish, or preparation of the manuscript.

**Competing interests:** The authors have declared that no competing interests exist.

in fetal hemodynamics (blood pressure and velocities) except for increased diastolic velocities in the AoI.

---

## Author summary

Coarctation of the aorta is a common congenital heart defect where part of the aorta is narrowed, usually at the aortic isthmus. While this condition can be critical in newborns and may require immediate surgery, fetuses generally do not show significant symptoms due to the presence of natural shunts that redirect blood flow and their ability to readapt, making early diagnosis challenging. In this study, we explored the conditions under which fetal circulation can compensate for this narrowing while maintaining normal function. Our findings suggest that for healthy blood flows and pressures to occur with aortic narrowing, the right ventricle must become larger than the left ventricle and the ductus arteriosus must widen. Additionally, we identified changes in blood flow velocity in the aortic isthmus that depended on the severity of the narrowing. These findings could help improve the early detection of aortic coarctation by recognizing these specific blood flow patterns during ultrasound examination.

## Introduction

Coarctation of the aorta (CoA) is one of the most prevalent congenital heart defects (CHD), accounting for 7–8% of all live births with CHD [1]. It is characterized by a narrowing of the aortic isthmus (AoI) and the aortic arch [2].

Postnatally, after ductal closure, a trans-coarctation pressure gradient is developed [3], and the CoA impairs left ventricular (LV) blood flow to the lower body. This increases afterload and pressure, potentially causing severe cardiac failure [4], which will also alter vessel wall properties if left untreated. Together with poor lower body perfusion, this becomes an important cause of morbidity and mortality. However, a prenatally CoA can rebalance the circulation due to the presence of the ductus arteriosus (DA). This shunt directs most of the right ventricular (RV) blood toward the aorta, bypassing the CoA site to reach the lower body and return blood to the placenta [2,5], thus preventing the formation of a hemodynamically important stenosis and a trans-coarctation pressure gradient. Additionally, fetuses can often remodel to compensate for adverse circulations. To prevent an increase in LV pressure that would result from forcing blood through the constricted AoI, the fetal cardiovascular system remodels so that most lower body flow is supplied by the RV through the DA [5], ensuring adequate delivery of blood to all organs. Therefore, two adaptive mechanisms occur: right ventricular disproportion and DA dilation.

However, the impact of these defects on CoA hemodynamics remains unclear, primarily because the fetoplacental circulation and adaptive mechanisms prevent major changes in Doppler velocity traces. Additionally, studying these effects is challenging

due to the lack of clear data on fetal CoA hemodynamics and the significant variability in CoA presentations among patients, resulting in suboptimal fetal-stage diagnoses. In this context, computational models arise as promising tools for analyzing hemodynamic changes associated with CoA. 0D computational models have been widely used in the assessment of the fetal cardiovascular system [6–14], representing it as a network of interconnected compartments that account for temporal variations of volumetric flow rate (VFR) and pressure, without considering spatial dimensions. They provide a non-invasive means to conduct in-silico interventions throughout the entire fetal body, which might not be feasible in clinical practice, and simulate different configurations of defects.

Our study aimed to understand the effect of CoA-related cardiovascular remodeling on fetal hemodynamics using a 0D computational model of fetal circulation.

## Materials and methods

### Ethics statement

The study protocol was approved by the Ethical Committee of the institution (Comité ético de investigación clínica del Hospital Clínic de Barcelona, Reg. HCB/2015/0365 and HCB/2019/0540) and written consent was obtained from all participants.

### Study population

We performed a retrospective analysis on 23 cases of antenatally diagnosed CoA, which were also confirmed postnatally. These 23 cases were classified as high risk for CoA antenatally based on established echocardiographic criteria: significant right dominance at the level of four-chamber and great vessels views (right-o-left ratio ≥ 1.4) and AoI diameter z-score < −2 at the three vessels and trachea or longitudinal aortic arch views evaluated in a tertiary referral center for fetal CHD (Hospital Clínic and Hospital Sant Joan de Déu) in Barcelona, Spain, from December 2014 to September 2018. Of the original 23 cases, only 9 had all images of acceptable quality for Doppler waveform extraction and cardiac morphometric assessment. These cases were examined between 27 and 39 (mean 32.10 ± 4.35) weeks of gestational age (GA). Additionally, we selected 7 cases from a prospective observational study at the same center that included singleton and uncomplicated spontaneously conceived pregnancies, without known maternal or gestational conditions potentially affecting cardiovascular remodeling. They were evaluated from February to April 2024 at the same GA as CoA cases (mean GA 31.80 ± 3.89 weeks, range between 28 and 37 weeks).

Fetal echocardiography was performed by experienced fetal medicine specialists (I.S. and P.V.) with Siemens Sonoline Antares (Siemens Medical Systems, Malvern, PA, USA) or Voluson E10 (General Electric, Zipf, Austria) machines with curved-array 2 − 6 MHz or 2 − 10 MHz transducers.

Extended fetal echocardiography was performed in all cases as per recommended guidelines [15–17], with a detailed assessment of the longitudinal aortic arch. In addition to fetal cardiac dopplers, estimated fetal weight (EFW) and doppler flows in the umbilical artery, middle cerebral artery, ductus venosus, and uterine artery were also assessed. The cardiac morphometric assessment included biventricular dimensions (ventricular areas) at a 4-chamber view. AoI and DA diameters with their respective z-scores [15,18] were measured at 3 vessel and trachea and/or longitudinal views, depending on the fetal position at acquisition. We computed the RV/LV end-diastolic volume (EDV) ratio using the RV/LV end-diastolic area ratio, assuming that the ventricular depths are comparable, resulting in very similar ratios. This assumption was necessary due to the difficulty of measuring ventricular depth in a fetus and the lack of prior studies indicating its variation. However, to validate this approach, we also conducted experiments considering ventricular depth as proportional to ventricular area variation (see S1 Appendix and S1 Fig). Ventricular dimensions were measured at end-diastole while great vessels were evaluated at mid-systole. Regarding cardiac flow evaluation, biventricular inflows were obtained in mid-diastole, placing the pulsed-wave Doppler sample immediately below the mitral and tricuspid valve leaflets. LV and

RV outflows, AoI and DA flows were obtained in mid-systole in the above-mentioned views. All fetoplacental flows were obtained with angles as close to 0º as possible and always below 30º. All areas, diameters, and velocities were manually delineated off-line in our Doppler image visualization platform [19,20].

## Statistical analysis

Data were analyzed using the SciPy package [21] in Python with Anaconda distribution (Anaconda Inc., Austin, TX, USA). The Shapiro-Wilk test of normality was conducted for continuous variables, which were reported as mean ± standard deviation for those with normal distributions and median (interquartile range) for non-normal distributions. Comparisons between the control and CoA cohorts were performed using Student's t-test or Mann-Whitney U-test, as appropriate. Categorical variables were expressed as n (%) and differences between the cohorts were compared using the chi-squared test. P-values < 0.05 were considered statistically significant.

## Computational model of the fetal circulation

**Construction of the model.** We built a 0D computational model of fetal circulation using *OpenCOR Version 0.6* [22] and *Python 3.9.12* [23]. Our equivalent electrical lumped model (Fig 1) represented the entire closed-loop circulation without external inputs or outputs. It encompassed the heart, modeled after Arts et al. [24], and the vasculature, based on Pennati *et al.* [6] and Garcia-Canadilla *et al.* [7,8], which included the great arteries and all vessels routinely evaluated on fetal ultrasound.

In our model, ventricles and atria were represented as single fiber chamber modules [24], cavities surrounded by fibrous walls whose mechanics were coupled to a representative sarcomere length. Myofiber stress correlated non-linearly to the average fiber extension, dependent on the cavity-to-wall volume ratio, enabling shape and size modifications. Aortic, pulmonary, mitral, and tricuspid valves were modeled as flow ducts, whose motion depended on the net instantaneous pressure difference approximated by the Bernoulli equation for smooth opening and closing [25].

Our fetal blood circulation model incorporated 44 compliant vascular compartments, combining vessel segments (arteries and veins) and vascular beds (organs). Each vessel segment included a resistance (due to blood viscosity), an inductor (blood inertia), and a capacitor (vessel compliance). Vascular beds consisted of two-element Windkessel models, composed of a resistance (organ peripheral resistance) and a capacitor (organ compliance).

To accurately represent the aortic arch, we defined four segments using established anatomical landmarks: ascending aorta, proximal and distal aortic arch, and AoI. We added a non-linear term $K$ representing the Bernoulli effect in the AoI compartment since stenosis behaves non-linearly [26]. The non-linear term satisfies the equation $\Delta p(t) = R \cdot Q(t) + K \cdot Q(t)^2 + L \cdot \frac{dQ(t)}{dt}$ (1), where $t$ is time, $p(t)$ the instantaneous local pressure, $Q(t)$ the VFR, $R$ the linear resistance, and $L$ the inductance. We modeled the lungs as a single vascular bed, as standardized in literature [27], and the upper body and brain as four distinct vascular beds distinguishing left and right sides. The lower body encompassed renal, hepatic, intestinal, lower extremities, and umbilico-placental circulations. Finally, we added fetal shunts (ductus venosus, foramen ovale, and DA) as vessel segments with an additional non-linear term accounting for the Bernoulli effect [6].

Each compartment in our model outputted pressure at the beginning and VFR at the end. Heart cavities additionally outputted cavity volume. Blood flow velocities were estimated as $v = Q/(\pi r^2)$ (2), with $Q$ being VFR and $r$ vessel radius. For the umbilical artery and middle cerebral artery, assuming VFR followed a laminar pattern, maximum velocity was defined as twice the average [28]. We acknowledged that most human fetuses present two umbilical arteries, while our simplified model combined them into one, considering half of the simulated blood flows through each.

The middle cerebral artery's VFR was estimated based on the flow from the left and right internal carotid arteries, assuming that 75% of their combined flow is directed to it [29]. To account for the exclusion of vertebral arteries in our

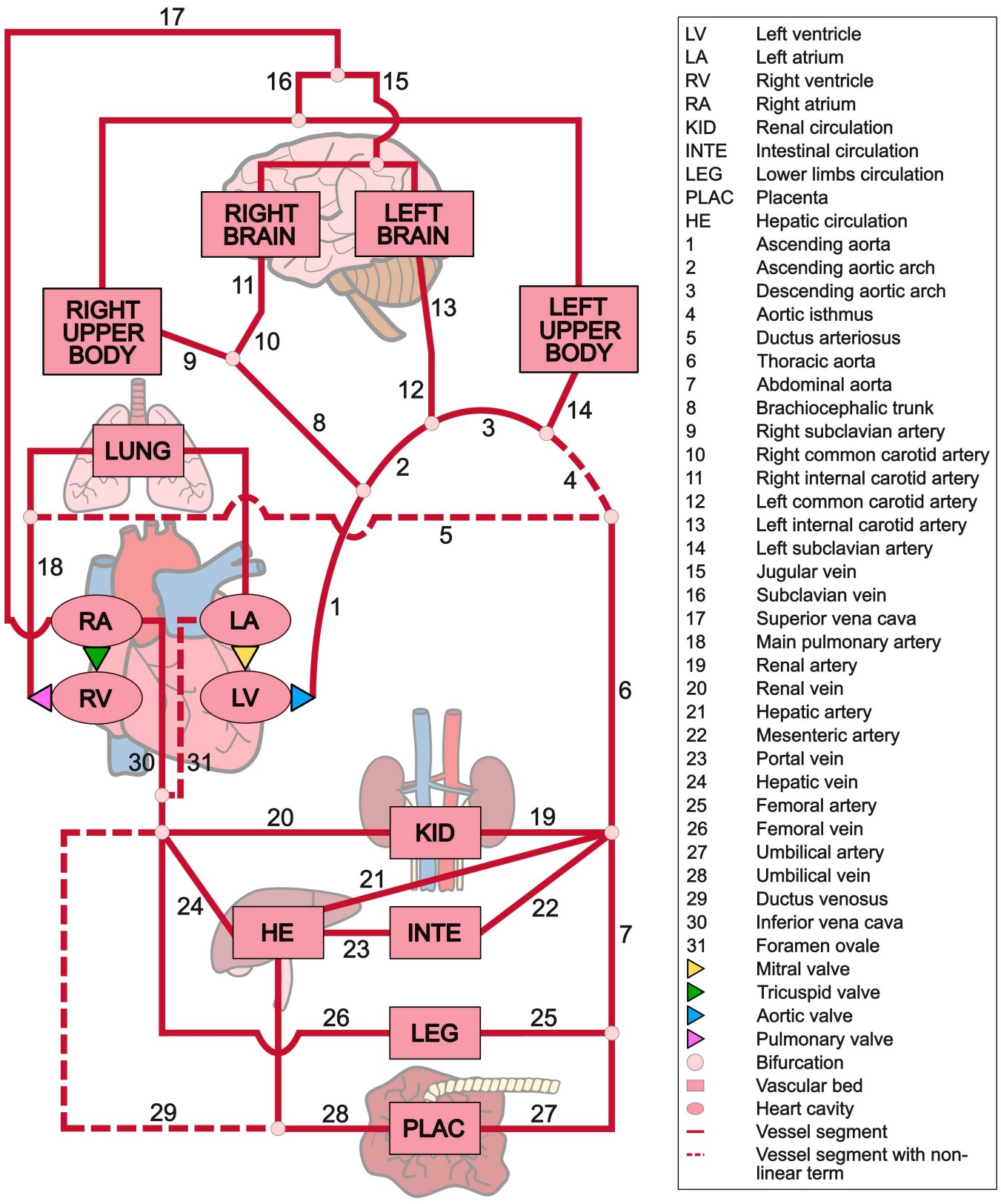

**Fig 1. Simplified anatomical configuration of the lumped computational model of fetoplacental circulation.**

model, a correction factor of 74.3% [30] was applied, representing the total cerebral blood supply provided by the middle cerebral artery. Finally, its pressure was calculated as the mean pressure of the left and right carotid arteries.

### Adjusting the model to healthy conditions

We adjusted model parameters with ultrasonographic data from 7 controls to obtain blood pressures, VFR, and velocities within physiological ranges for a 32-week gestation fetus (see black curves and circles in Figs 2-4). Ventricular and atrial parameters were manually adjusted and those from vascular beds and vessel segments were computed using the Pennati and Fumero equations [31] (see S1 Appendix). We compared simulated blood flow velocities with Doppler velocity traces from control cases, aligning cardiac cycle times.

### Simulation of coarctation of the aorta

To adequately represent the hemodynamics of a fetus with CoA, we incorporated the published cardiovascular pattern associated with CoA, consisting of AoI narrowing with right ventricular disproportion and DA dilation. We simulated various degrees of CoA based on literature-derived ranges.

**CoA abnormalities in the literature.** We extracted from literature the greatest variation in cardiac parameters for CoA compared to control populations (see S1 Appendix, and S1 and S2 Tables). Subsequently, we defined our model's ranges slightly beyond the reported values to include potential extreme cases not documented in literature: 100%-20% AoI diameter, 100%-150% DA diameter, and 1.05-5 RV/LV EDV ratio.

**Simulation of CoA abnormalities.** The following modifications we introduced in our model:

**Aortic narrowing:** We decreased the AoI diameter up to 20% of its reference value. To ensure that the blood flow was achieved with minimal biological work, we recalculated other aortic arch segments' diameters using Murray's law [32], which dictates that when a vessel with a radius $r$ bifurcates in two branches with radii $r_1$ and $r_2$, the three radii must satisfy the following equation: $r^3 = r_1^3 + r_2^3$ (3). The resistances ($R$), capacitances ($C$) and inductances ($L$) of the affected vessel segments were recalculated according to the resulting radius [7]. Using the equations $R = \frac{8\mu l}{(\pi r^4)}$ (4), $C = \frac{3\pi r^3}{(2Eh)}$ (5), $L = \frac{\rho l}{\pi r^2}$ (6), where $r$ and $l$ are the radius and length of the segment, $h$ is its wall thickness (assumed to be 15% of $r$ [14]), $\mu$ is blood viscosity and $E$ is Young's modulus, we assumed $R \propto r^{-4}$, $C \propto r^2$ and $L \propto r^{-2}$. We considered that adaptation of the cerebral branches due to the aortic narrowing was negligible compared to other associated CoA defects [33].

**Ventricular disproportion:** We increased the RV/LV EDV ratio up to 5 by decreasing the LV size and enlarging the RV. The dimensions of atria and valves we adjusted accordingly. Details on how these changes were applied are given in S1 Appendix.

**Ductus arteriosus dilation:** We increased the DA diameter up to 150% of its reference value, recalculating its electrical components according to the resulting radius [7] as was performed for *Aortic narrowing*.

### Evaluation of the CoA simulations

We assessed all CoA simulations to determine cardiovascular patterns consistent with physiological data, hypothesizing that organ perfusion, wall shear stresses, and pressures throughout the entire fetal circulation must resemble those in healthy cases to avoid circulatory adaptations. We computed a series of indices and defined physiological regions where they deviated less than 15% (arbitrarily chosen) from the control fetuses' mean values. First, we calculated total blood flow volume as the time-integral of VFR along a cardiac cycle. Then, we estimated maximum wall shear stress in a vessel as the maximum of $4\mu Q/r^3$ (7), with $Q$ being VFR, $\mu$ blood viscosity, and $r$ vessel radius. Finally, the maximum pressure along a cardiac cycle was determined. To sum up:

1. We estimated the total blood flow reaching different fetal organs, including the thoracic and abdominal aorta, the middle cerebral artery, the superior vena cava, the hepatic artery, the renal artery, the umbilical artery, the inferior vena cava, and the lungs.

 

2. We assessed the maximum wall shear stress in the aforementioned vessels and included the ductus arteriosus, main pulmonary artery and ascending aorta.

3. Finally, we obtained the maximum pressure in all previous vessels and also in all the cavities of the heart.

We computed the percentage changes in total blood flow volume, maximum wall shear stress, and maximum pressure compared to our reference healthy model (see S2 Fig). We defined physiological regions (see Fig 3) as combinations of AoI narrowing, RV/LV EDV ratio, and DA dilation with changes below 5% (dark green), 10% (medium green), and 15% (light green) across all anatomical sites.

### Comparison with real data

To validate our results, we mapped real healthy and CoA cases (grey and white crosses in Fig 3) to verify if they were placed within physiological regions. We defined AoI narrowing and DA dilation as the percentage of variation of ultrasound-estimated diameters compared to the population mean for that GA [34,35]. Ventricular disproportion was computed as RV/LV end-diastolic area ratio, assuming that ventricular depth does not change significantly, resulting in a ratio comparable to that of EDV.

### Qualitative assessment of simulated waveforms

For 100%, 80%, 60%, and 40% of the reference AoI size, we identified the combination of ventricular disproportion and DA dilation that provided the minimum variation from our reference healthy model, computed as the centroids of physiological regions below 10% of percentual change (circles in Fig 3). Their blood pressures, velocities, and VFR in heart valves, DA, AoI, middle cerebral artery, and umbilical artery were plotted to analyze their patterns (see Fig 4).

## Results

### Characteristics of the study population

We summarized the perinatal characteristics of the study population in Table 1. The control and CoA groups had similar mean maternal age, weight and height, GA at scan and delivery, EFW, and birth weight. As expected, CoA cases showed lower AoI diameter and RV/LV EDV ratio, with a tendency towards larger DA than controls. No significant differences were observed in AoI, DA, umbilical artery, and middle cerebral artery pulsatility indices. The complete dataset for each control and CoA case is available in S1 Data.

### Validation of the reference healthy model

Our simulated healthy blood flow velocity waveforms closely resembled the Doppler velocity traces obtained from controls (Fig 2). Our simulations produced LV and RV ejection times of 35.0% and 35.2% of the cardiac cycle and filling times of 41.3% for both ventricles. In comparison, real control data indicated mean ejection times of 36.7% (33.5%-41.0%) for LV and 39.0% (36.0%-41.4%) for RV, and filling times of 43.1% (38.6%- 47.0%) for LV and 38.6% (34.7% to 42.9%) for RV. Therefore, there were no significant changes in cardiac-cycle timing parameters. The velocity waveforms from the control cases in our cohort are available in S2 Data.

### Simulations of aortic coarctation

**Comparison with real data.** In CoA scenarios, physiological regions (green areas in Fig 3) emerged at small RV/LV EDV ratio (1.05-1.67) and low DA dilation (100%-115% of reference DA diameter) for mild AoI narrowing (100%- 80% of reference AoI diameter). For severe AoI narrowing (40–20% of reference AoI diameter), maintaining physiological hemodynamics required higher RV/LV EDV ratio (2–5) and greater DA dilation (105%-125% of reference DA diameter). In

**Table 1. Maternal and fetoplacental characteristics of the study population.**

| Parameter | Controls (n = 7) | CoA (n = 9) | p-value |
|---|---|---|---|
| Maternal age (years) | 33.98 ± 6.98 | 33.62 ± 3.74 | 0.911 |
| Maternal weight (kg) | 68.40 ± 15.26 | 61.75 ± 4.57 | 0.433 |
| Maternal height (m) | 1.62 ± 0.06 | 1.67 ± 0.06 | 0.205 |
| Gestational age at scan (weeks) | 31.80 ± 2.89 | 32.10 ± 4.35 | 0.878 |
| Estimated fetal weight (g) | 1832.86 ± 566.80 | 1946.67 ± 820.08 | 0.759 |
| Aortic isthmus diameter z-score | 0.12 ± 0.57 | -2.11 ± 1.47 | 0.002 |
| Ductus arteriosus diameter z-score | -0.08 ± 0.20 | 0.19 ± 0.34 | 0.086 |
| RV/LV end-diastolic volume ratio | 1.11 ± 0.05 | 1.63 ± 0.12 | 0.002 |
| Aortic isthmus pulsatility index | 3.63 ± 0.87 | 3.85 ± 0.58 | 0.617 |
| Ductus arteriosus pulsatility index | 2.81 ± 0.93 | 2.84 ± 0.28 | 0.954 |
| Umbilical artery pulsatility index | 0.97 ± 0.15 | 1.19 ± 0.46 | 0.244 |
| Middle cerebral artery pulsatility index | 1.90 ± 0.33 | 1.91 ± 0.57 | 0.983 |
| Vaginal delivery | 4 (57.14) | 6 (66.67) | 1.000 |
| Gestational age at birth (weeks) | 40.10 (39.25 − 40.80) | 40.00 (38.57 − 40.00) | 0.426 |
| Birth weight (g) | 3201.86 ± 390.68 | 3187.78 ± 682.00 | 0.961 |
| 5-min Apgar score < 7 | 1 (14.29) | 0 (0) | 0.896 |

Data are expressed as mean ± standard deviation, median (interquartile range), or n (%). CoA: coarctation of the aorta, g: grams, kg: kilograms, LV: left ventricle, m: meters, mm: millimeters, RV: right ventricle.

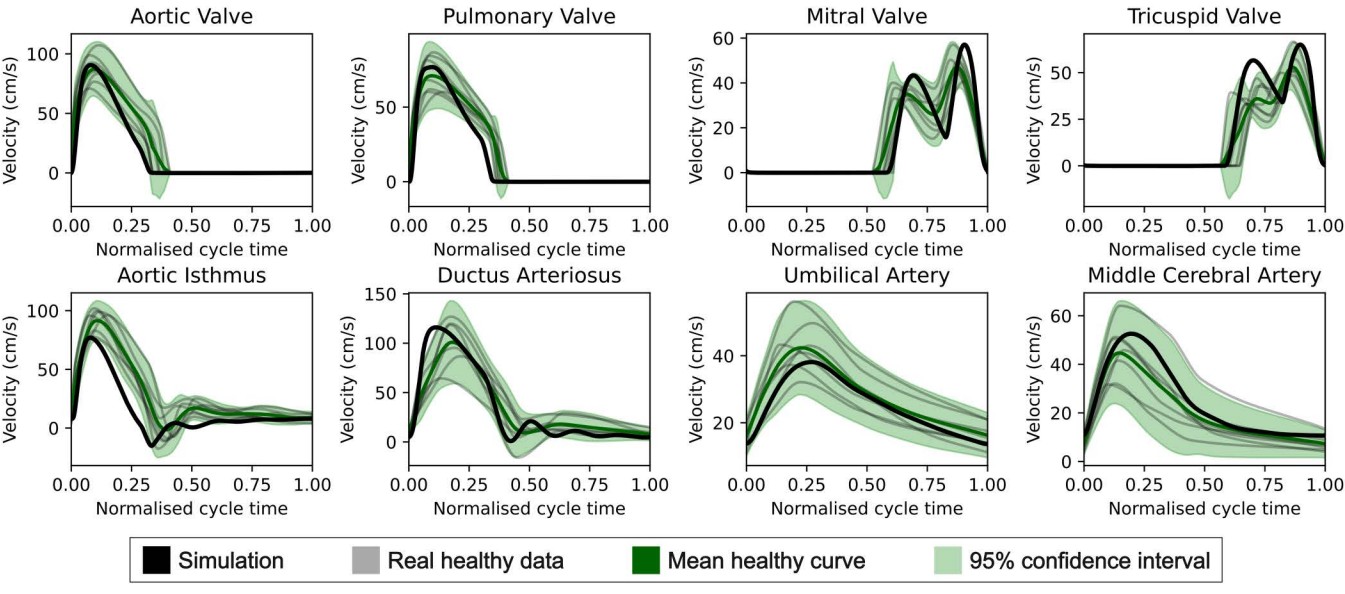

**Fig 2. Comparison of the computational model's healthy baseline simulated traces (black) with those obtained by ultrasound from control cases (grey), after manually tuning the model's parameters to closely align with the confidence intervals (green).**

extreme cases (20% AoI), scenarios without ventricular disproportion resulted in increased LV pressure, increased total blood flow volume in the middle cerebral artery, and decreased wall shear stress in the ascending aorta (S2A-S2C Fig). Conversely, without DA dilation wall shear stress in the DA increased (S2D Fig). All real controls and CoA cases (grey

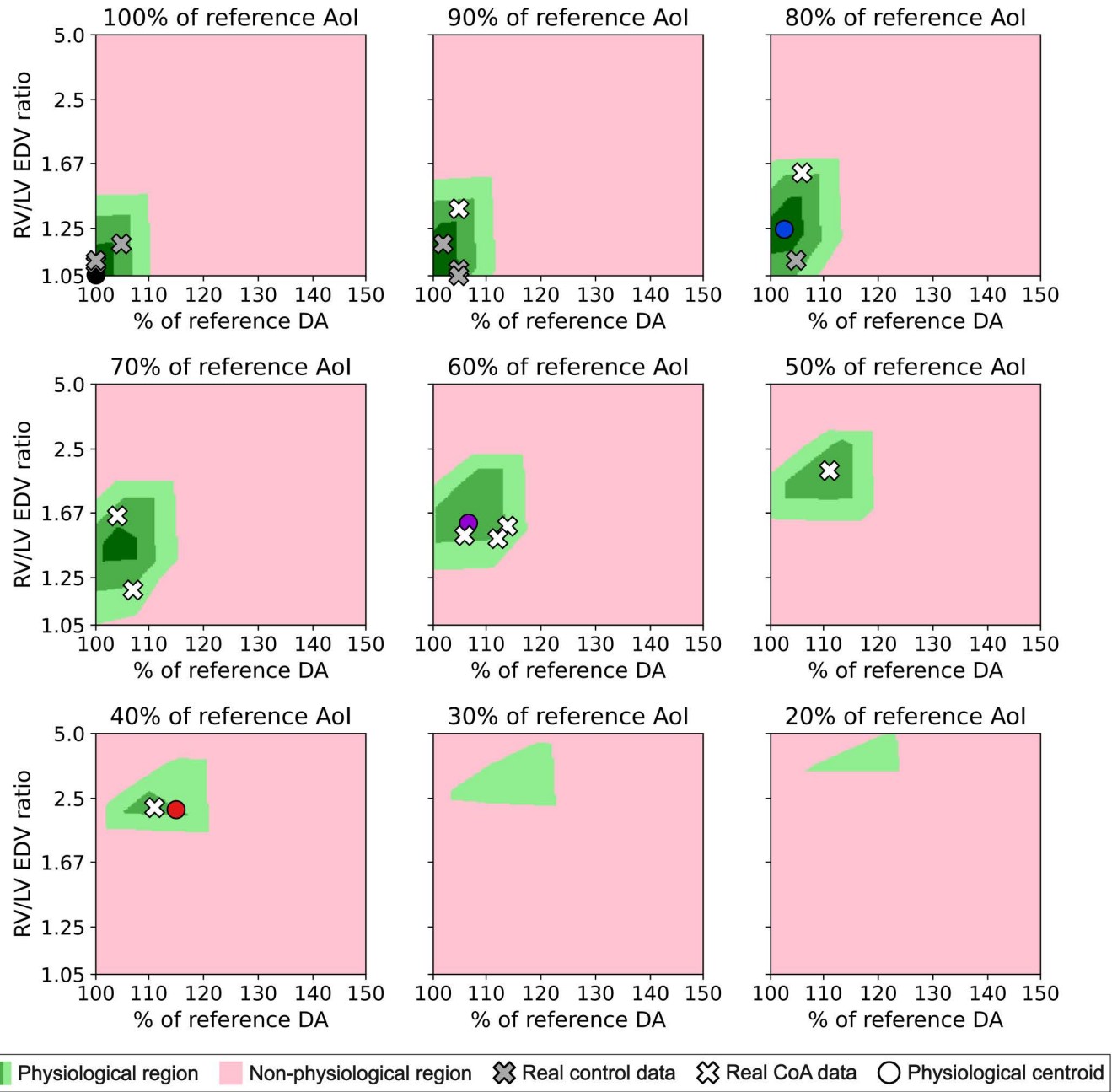

**Fig 3. Distribution of regions in which total blood flow, wall shear stress and pressures are below a 5% (dark green), 10% (medium green), and 15% (light green) change in the whole fetal body relative to the healthy values, or exceed that amount at any anatomical site (pink), for different combinations of aortic isthmus (AoI) narrowing, right-to-left ventricular end-diastolic volume (RV/LV EDV) ratio and ductus arteriosus (DA) dilation.** Real healthy and coarctation of the aorta (CoA) cases are marked with grey and white crosses, respectively. Points in which the percentage of change of the physiological region with respect to the healthy reference model is minimal are marked with grey, blue, purple, and red circles for 100%, 80%, 60%, and 40% of the reference aortic isthmus size, respectively.

and white crosses) were located within physiological regions in all AoI narrowing degrees. Confidence intervals for the physiological regions are given in S1 Appendix and S3 Fig.

**Comparison of different degrees of CoA.** Fig 4 depicts blood pressure, velocity, and VFR curves simulated for RV/LV EDV ratio and DA dilation combinations minimizing changes from the reference healthy model. Pressure levels remained stable across simulations. Minor changes occurred in the umbilical artery, pulmonary, tricuspid, and mitral velocities; a slight decrease in aortic valve velocity; and a slight increase in the middle cerebral artery and DA velocities. Strikingly, AoI flow velocity considerably increased from 75 cm/s in the healthy case to 125 cm/s in 40% AoI narrowing. VFR decreased through AoI and mitral and aortic valves, increased through DA and pulmonary and tricuspid valves, and remained stable through the middle cerebral artery and umbilical artery.

## Discussion

We developed a closed computational model of fetal circulation to study hemodynamic changes associated with CoA. It allowed us to simulate the most common cardiovascular patterns associated with fetal CoA, with different degrees of AoI narrowing, ventricular disproportion, and DA dilation. Thus, we could identify the specific combinations that allow to maintain physiological circulation in terms of blood pressure, velocity, and volume at different anatomical sites, as observed in clinical data. To the best of our knowledge, this is the first model to successfully simulate all fetal CoA defects.

We first qualitatively validated our reference model, representing a healthy fetus of 32 weeks of GA, by comparing the simulated blood velocity traces with those obtained from echocardiographic images in healthy controls at a similar GA. Visually (see Fig 2), all simulated curves matched the normality ranges of the control population, falling within or very close to the 95% confidence interval regions. The most notable deviation was observed in the aortic isthmus velocity, which is expected given the challenges in accurately measuring this vessel. Since our model simulates blood flow, converting it to velocity can introduce errors due to uncertainties in both vessel diameter and the spatial flow profile. Moreover, our cases cover a gestational age range of 6 weeks, but considering the small changes in pulsatility indices during the third trimester of pregnancy [36], we assumed that gestational age-related hemodynamic differences were negligible. The obtained LV and RV ejection and filling times remained within physiological limits [37], although ejection times were discretely shortened.

The simulations combining different degrees of AoI narrowing showed the necessity of co-occurrence of ventricular disproportion and DA dilation to compensate for the coarctation [38–40]. Without ventricular disproportion, extreme aortic narrowing would cause increased intraventricular pressure regardless of DA dilation and potentially harm the heart, leading to overperfusion of the brain and impaired development of the lower body and the placenta (S2A-S2B Fig). This is consistent with Giménez *et al.* [41], who showed that ventricular disproportion normalized LV pressures. However, severe coarctations will probably present additional defects, such as bicuspid aortic valve, ventricular septal defect, or mitral valve disorders [42], that we have not considered in our study since our database only contained isolated CoA. Additionally, wall shear stress in the ascending aorta and DA (S2C-S2D Fig) showed that the degree of ventricular disproportion and DA dilation, respectively, appeared to be dependent on AoI narrowing. Both the healthy and CoA real data mapped onto Fig 3 supported our findings, as they consistently were located within the defined physiological regions, even for severe AoI narrowing. While fetal imaging measurements inherently contain some error, the real data still were located within or very close to physiological regions when accounting for ventricular depth variations proportional to ventricular area changes (see S1 Appendix and S1 Fig). Additionally, the data remained within confidence intervals (see S1 Appendix and S3 Fig), further reinforcing the validity of our defined physiological regions while accounting for the expected variability within a population. In cases where the combination of DA dilation and ventricular disproportion failed to compensate for aortic narrowing (pink regions in Fig 3), the cardiovascular system would remodel to ensure adequate perfusion of major organs. This adaptation could involve adjusting either of these factors (ventricular disproportion, DA dilation, and especially AoI narrowing) so that the system approaches the physiological region, or lead to the onset of additional defects.

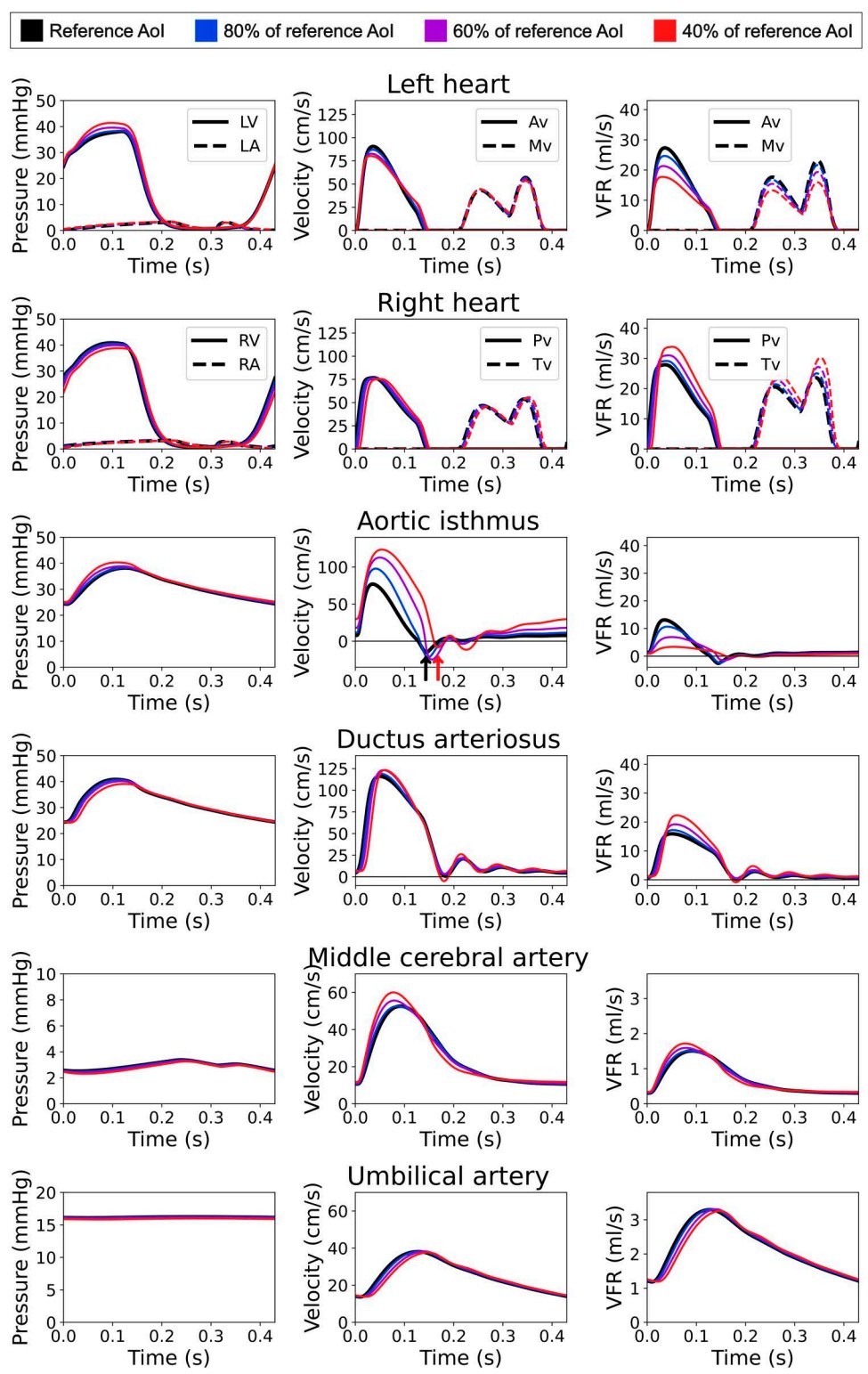

**Fig 4. Blood pressure, velocity, and volumetric flow rate (VFR) curves at left heart (LV: left ventricle, LA: left atrium, Av: aortic valve, Mv: mitral valve), right heart (RV: right ventricle, RA: right atrium, Pv: pulmonary valve, Tv: tricuspid valve), ductus arteriosus, aortic isthmus (AoI), middle cerebral artery, and umbilical artery obtained with the computational model at 100% (black), 80% (blue), 60% (purple) and 40%**

**(red) of the reference aortic isthmus diameter size, for the combination of right-to-left ventricular end-diastolic volume ratio and ductus arteriosus dilation with minimum percentage of change compared to the healthy baseline.** Arrows in the aortic isthmus velocity indicate the onset of the reversed peak for the reference model and 40% of the reference aortic isthmus size.

Finally, we demonstrated that blood pressure and velocities at different fetal anatomical sites (first 2 columns of Fig 4) present minimal changes compared to those of a healthy fetus regardless of the severity of coarctation. This differs from the work of Chen *et al.* [43], which found decreased AoI pressure and highly increased AoI velocity, possibly because their open model imposed healthy flows as boundary conditions and did not include cardiovascular adaptations associated with AoI narrowing. We also observed significant differences in AoI blood velocity, which increased but within normal physiological ranges, consistent with clinical findings [44]. Additionally, the velocity pattern in AoI showed a delayed retrograde flow peak, possibly due to the longer RV ejection compared to LV, although due to data variability (see Fig 2) accurately measuring this in clinical practice would be challenging. AoI velocity also presented increased antegrade diastolic velocity, because despite the small volume of blood passing through the AoI in diastole, it was narrower. Diastolic flow persistence at AoI, known postnatally as diastolic run-off [45], has been reported in clinical fetal studies [46–48] and could potentially be used to improve the accuracy of CoA diagnosis (S4 and S5 Figs). Changes were also observed in volumetric flow rate: it decreased through the AoI and the mitral and aortic valves, while through the middle cerebral artery and umbilical artery remained similar to that in a healthy fetus, consistent with Giménez *et al.* [41]. Unlike in healthy cases, where the lower body and the placenta are supplied by both ventricles with a higher RV contribution, in severe CoA they are almost exclusively supplied by the RV via DA, as the LV output mainly perfuses the upper body with minimal flow arriving to the AoI. Therefore, we corroborated that in fetal CoA a redistribution of blood flow occurs to ensure perfusion of the brain and placenta, without a significant alteration in fetal hemodynamics (blood pressure and velocities) with the exception of increased diastolic velocities in the AoI.

0D models have some limitations. Firstly, 0D lumped models do not consider wave reflections or spatial variation of variables (pressure, volumetric flow rate, and volume) within a component. However, in our study we were interested in the overall distribution of flow, and there were moderate Reynolds numbers (below 2300), which support our assumption of laminar flow. Additionally, while the 0D lumped model did not allow for a precise assessment of localized wall shear stresses across the fetal circulation, our approximation was sufficient for analyzing proportional changes relative to the healthy scenario. Therefore, we believe that a 0D lumped model was sufficiently accurate for our purposes. Additionally, our model simplified the cardiovascular system, as it only included one umbilical artery and a single component for the lungs and lacked a specific component for the middle cerebral artery. Despite these simplifications, we consider the model to be adequately accurate, as all velocities could be estimated from existing model components, as shown in Fig 2. Furthermore, our study was based on a small cohort, due to the limited number of CoA cases detected at the fetal stage that also had complete, high-quality images available for analysis. Expanding the dataset with a wider range of fetuses at different GA would be beneficial to improve the model's robustness and generalizability and to allow for direct comparisons between real and simulated CoA velocity traces. Personalizing our model with different CoA cases within the clinical spectrum of the complex CHD would also be valuable to observe specific changes compared to healthy cases. Finally, although our CoA simulations showed increased antegrade diastolic velocity in the aortic isthmus, there is variability in the velocity patterns among patients. Future research should focus on identifying a more reliable diagnostic biomarker for this pathology.

In conclusion, our closed 0D computational model of fetal circulation is a good approximation to assess hemodynamic changes in fetuses with CoA. For the first time in literature, we effectively simulated the hemodynamics of CoA, capturing the interactions between AoI narrowing, ventricular disproportion, and DA dilation. Our simulations compensate for CoA, with physiological regions varying according to the severity of aortic narrowing. These findings were supported by clinical

data from a series composed of control and CoA cases, confirming that blood flow redistribution ensures adequate organ perfusion without notably affecting blood pressures and velocities, except for the AoI velocity pattern, which might aid in diagnosing CoA. Future studies should expand the dataset and incorporate patient-specific CoA data to further enhance the model's applicability.

## Supporting information

**S1 Appendix. Supporting information.**
(DOCX)

**S1 Data. Clinical data of the control and CoA cases in our cohort.**
(XLSX)

**S2 Data. Velocity tracings of the healthy controls in our cohort.)**
(CSV)

**S1 Table. Aortic isthmus diameter reported in different clinical studies and expected greatest variation in coarctation of the aorta with respect to control populations.**
(DOCX)

**S2 Table. Ductus arteriosus diameter reported in different clinical studies and expected greatest variation in coarctation of the aorta with respect to control populations.**
(DOCX)

**S1 Fig. Physiological regions considering ventricular depth variations.** Distribution of regions in which total blood flow, wall shear stress and pressures are below a 5% (dark green), 10% (medium green), and 15% (light green) change in the whole fetal body relative to the healthy values, or exceed that amount at any anatomical site (pink), for different combinations of aortic isthmus (AoI) narrowing, right-to-left ventricular end-diastolic volume (RV/LV EDV) ratio and ductus arteriosus (DA) dilation. Real healthy and coarctation of the aorta (CoA) cases are marked with grey and white crosses, respectively, where ventricular depth is assumed to change proportionally to the ventricular area. Points in which the percentage of change of the physiological region with respect to the healthy reference model is minimal are marked with grey, blue, purple, and red circles for 100%, 80%, 60%, and 40% of the reference aortic isthmus size, respectively.
(TIFF)

**S2 Fig. Most important percentage changes in CoA scenarios.** Percentages of change of: total blood flow through (A) the middle cerebral artery and (B) the umbilical artery; wall shear stress at (C) ascending aorta, (D) main pulmonary artery, (E) ductus arteriosus (DA); and (F) pressure at the left ventricle for different combinations of aortic isthmus (AoI) narrowing, right-to-left ventricular end-diastolic volume (EDV) ratio and DA dilation. Isolines indicate absolute change percentages of 0%, 5%, 10% and 15%.
(TIFF)

**S3 Fig. Confidence intervals for physiological regions.** Distribution of regions in which total blood flow, wall shear stress and pressures are below a 5% (dark green), 10% (medium green), and 15% (light green) change in the whole fetal body relative to the healthy values, or exceed that amount at any anatomical site (pink), for different combinations of aortic isthmus (AoI) narrowing, right-to-left ventricular end-diastolic volume (RV/LV EDV) ratio and ductus arteriosus (DA) dilation. 25% (blue) and 75% (orange) confidence interval areas are also depicted.
(TIFF)

**S4 Fig. Normal fetus at 32 weeks of gestation.** (A) The 3-vessel and trachea view evaluated with conventional color Doppler shows synchronic anterograde flow in the transverse aortic arch and aortic isthmus compared with the main pulmonary artery and ductus arteriosus. (B) A small amount of reversed flow is evident at the end of the systole (codified in blue). (C) Aortic isthmus flow characteristics by spectral pulsed-Doppler. Note the presence of a late-systolic notch (yellow arrow) with a small reversal flow wave (*) and very low flow rate in diastole (white arrows). Ao: aorta, DA: ductus arteriosus, Sp: spine. (TIFF)

**S5 Fig. Confirmed aortic coarctation case at 32 weeks of gestation.** (A) Color Doppler 3-vessel and trachea view showing persistence of diastolic flow in the transverse aortic arch and aortic isthmus compared with the main pulmonary artery and ductus arteriosus. B) Longitudinal aortic arch view evaluated with color Doppler. (C) Aortic isthmus flow characteristics by spectral pulsed-Doppler. Note the presence of a late-systolic notch (yellow arrow) with a small reversal flow wave (*) and clearly increased diastolic flow (white arrows). Ao: aorta, DA: ductus arteriosus, Sp: spine. **Left subclavian artery. (TIFF)

## Acknowledgments

We thank all study participants for their personal time and commitment to this project. We also thank Laia Grau from BCNatal Fetal Medicine Research Center (Hospital Clínic and Hospital Sant Joan de Déu, Universitat de Barcelona, Barcelona, Spain) for her assistance in the literature search on aortic isthmus flow in fetal life.

## Author contributions

**Conceptualization:** Inmaculada Villanueva-Baxarias, Anna Pellisé-Tintoré, Patricia Garcia-Canadilla, Oscar Camara, Bart Bijnens, Gabriel Bernardino.

**Data curation:** Inmaculada Villanueva-Baxarias, María Pérez-Rodríguez, Laura Nogué, Pooja Vaziraani, Iris Soveral, Fàtima Crispi, Olga Gómez.

**Formal analysis:** Inmaculada Villanueva-Baxarias, María Pérez-Rodríguez.

**Investigation:** Laura Nogué, Pooja Vaziraani, Iris Soveral, Fàtima Crispi, Olga Gómez.

**Methodology:** Inmaculada Villanueva-Baxarias, Anna Pellisé-Tintoré, Patricia Garcia-Canadilla, Oscar Camara, Bart Bijnens, Gabriel Bernardino.

**Resources:** Laura Nogué, Pooja Vaziraani, Iris Soveral, Fàtima Crispi, Olga Gómez.

**Software:** Inmaculada Villanueva-Baxarias, Anna Pellisé-Tintoré, Gabriel Bernardino.

**Supervision:** Patricia Garcia-Canadilla, Oscar Camara, Bart Bijnens, Gabriel Bernardino.

**Validation:** Inmaculada Villanueva-Baxarias.

**Visualization:** Inmaculada Villanueva-Baxarias, Patricia Garcia-Canadilla.

**Writing – original draft:** Inmaculada Villanueva-Baxarias.

**Writing – review & editing:** Inmaculada Villanueva-Baxarias, Laura Nogué, Pooja Vaziraani, Iris Soveral, Fàtima Crispi, Olga Gómez, Patricia Garcia-Canadilla, Oscar Camara, Bart Bijnens, Gabriel Bernardino.

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
