## [Decision Letter · Decision Letter 0]

9 Jan 2025

Understanding the hemodynamic changes in fetuses with coarctation of the aorta using a lumped model of fetal circulation

PLOS Computational Biology

Dear Dr. Villanueva Baxarias,

Thank you for submitting your manuscript to PLOS Computational Biology. After careful consideration, we feel that it has merit but does not fully meet PLOS Computational Biology's publication criteria as it currently stands. Therefore, we invite you to submit a revised version of the manuscript that addresses the points raised during the review process.

Please submit your revised manuscript within 60 days Mar 11 2025 11:59PM. If you will need more time than this to complete your revisions, please reply to this message or contact the journal office at ploscompbiol@plos.org. Please include the following items when submitting your revised manuscript:

We look forward to receiving your revised manuscript.

Kind regards,

Andrew D. McCulloch, Ph.D.

Academic Editor

PLOS Computational Biology

Marc Birtwistle

Section Editor

PLOS Computational Biology

**Journal Requirements:**

At this stage, the following Authors/Authors require contributions: Maria Inmaculada Villanueva Baxarias, Anna Pellisé-Tintoré, María Pérez, Laura Nogué, Pooja Vaziraani, Iris Soveral, Fàtima Crispi, Olga Gómez, Patricia Garcia-Canadilla, Oscar Camara, Bart Bijnens, and Gabriel Bernardino. Please ensure that the full contributions of each author are acknowledged in the "Add/Edit/Remove Authors" section of our submission form.

2) Thank you for including an Ethics Statement for your study. Please include:

i) The full name(s) of the Institutional Review Board(s) or Ethics Committee(s).

3) Please upload a copy of Fig D.1 which you refer to in your text on page 19. Or, if the figure is no longer to be included as part of the submission please remove all reference to it within the text.

4) We have noticed that you have uploaded Supporting Information files, but you have not included a complete list of legends. Please add a full list of legends for Appendix S1-S3 after the references list.

5) We notice that your supplementary figures are uploaded with the file type 'Figure'. Please amend the file type to 'Supporting Information'. Please ensure that each Supporting Information file has a legend listed in the manuscript after the references list.

Potential Copyright Issues:

i) Figure 4. Please confirm whether you drew the images / clip-art within the figure panels by hand. If you did not draw the images, please provide (a) a link to the source of the images or icons and their license / terms of use; or (b) written permission from the copyright holder to publish the images or icons under our CC BY 4.0 license. Alternatively, you may replace the images with open source alternatives. See these open source resources you may use to replace images / clip-art:

7) In the online submission form, you indicated that "The data that support the findings of this study are available on request from the corresponding author. All PLOS journals now require all data underlying the findings described in their manuscript to be freely available to other researchers, either

1. In a public repository

2. Within the manuscript itself

3. Uploaded as supplementary information.

8) In the online submission form you indicate that your data is not available for proprietary reasons and have provided a contact point for accessing this data. Please note that your current contact point is a co-author on this manuscript. According to our Data Policy, the contact point must not be an author on the manuscript and must be an institutional contact, ideally not an individual. Please revise your data statement to a non-author institutional point of contact, such as a data access or ethics committee, and send this to us via return email. Please also include contact information for the third party organization, and please include the full citation of where the data can be found.

9) Please amend your detailed Financial Disclosure statement. This is published with the article. It must therefore be completed in full sentences and contain the exact wording you wish to be published.

10) Please ensure that the funders and grant numbers match between the Financial Disclosure field and the Funding Information tab in your submission form. Note that the funders must be provided in the same order in both places as well. Currently, this funder "European Social Fund Plus (FSE+)" is missing from the Funding Information tab. In addition, the order of the grants is different in both places.

Please indicate by return email the full and correct funding information for your study and confirm the order in which funding contributions should appear. Please be sure to indicate whether the funders played any role in the study design, data collection and analysis, decision to publish, or preparation of the manuscript.

11) Population_characteristics.xlsx and Healthy_velocities.csv are currently uploaded as file type “Other”, which is not viewable by the reviewers. Please change the file type(s) to 'Supporting Information' and include a legend in the manuscript if you wish it/them to be included in review.

**Reviewers' comments:**

Reviewer's Responses to Questions

Reviewer #1: The manuscript applies an established lumped parameter model of the fetal cardiovascular system to explore the impact of geometric perturbations in the aortic isthmus (AoI), which is associated with coarctation of the aorta. The in silico model predicts certain robustness and adaptability of the fetal circulation to this pathophysiological condition and proposes the AoI diastolic velocity as a novel diagnostic indicator.

The authors are established experts in the field, and the manuscript is generally well-written and gives a good example of mechanistic insight into fetal pathophysiology supported by clinical data. Nonetheless, there are a few issues that should be addressed as suggested below.

Firstly, you state (p. 11) that “interactions between cardiovascular components without flow phenomena or wave reflections, [...] was sufficiently accurate for our purposes.”

Aren’t you concerned about the impact of potentially significant pulse wave reflections from the narrowed regions of the aorta? (see Cahill et al. eBioMed. 67, 103326 (2021) 10.1016/j.ebiom.2021.103326). Since the model deals with high Reynolds flows in the complex geometries of the aorta, would it be more appropriate to use distributed flow networks and/or hybrid 0D-1D-3D approaches (for example, Guettouche et al. [9] quoted by the authors, and also Wong et al. Ann. Biomed. Eng. 50, 1158 (2022) 10.1007/s10439-022-02990-5)? Please provide more evidence that these effects can be neglected.

Secondly, I am a bit surprised that the authors did not consult a highly relevant 2015 study of coarctication biomechanics by Zahra Keshavarz-Motamed and others (J. Biomech. 48, 4229,

10.1016/j.jbiomech.2015.10.017). In particular, I invite you to discuss Keshavarz-Motamed et al.’s observation that the trans-coarctation pressure drop is highly sensitive to compliance as well as cardiac output. Thus, the AoI geometry alone could be insufficient to explain the mechanisms of this condition.

Finally, although the model is based on prior work by the authors, I would expect further details on the model’s sensitivity to key parameters and comments on the associated uncertainty (please see examples in Marquis Math. Biosci. 304, 9 (2018) 10.1016/j.mbs.2018.07.001, and Mihaela J. R. Soc. Interface. 1720200886 (2020) 10.1098/rsif.2020.0886). This would increase the confidence in the model’s predictions and minimize the risk of overfitting to limited data.

Specific points

The Figures should be more clearly annotated, and the order could be confusing. For example, starting with Figure 4 and indicating the model parameters, which are being fitted and/or perturbed, would put the rest of the paper into proper context for the readership of this journal.

Figure 1 caption should be extended to explain how the baseline model was parametrized. Considering large variation in the measured control group, it would help to add a mean curve and shaded confidence intervals to aid the assessment. Furthermore, could you factor in the expected dispersion in the normal population when discussing the effect of altered aortic geometry and other CV parameters in Figs 2 & 3?

Page 11, l. 212. Please clarify what you mean by “blood pressure velocities”? There is no wave speed present in your model as far as I am aware.

Figure S3 (c) suggests that both min and max velocities are elevated. Does your model predict that the PI or RI indices are similar to the reference control? If so, what dimensionless index would you propose to detect this condition? Please also see the general point 2 above, which refers to the driving pressure gradient.

I recommend that the codes underpinning the modeled scenarios be made publicly available. Any personally identifiable data should be removed or replaced with aggregated or synthetic illustrations. Sufficient metadata should accompany the codes to allow for the reproduction of key figures and results.

Reviewer #2: Villanueva et al. present a study that develops a 0D mathematical model to simulate coarctation of the aorta (CoA) remodeling patterns. The study finds that mild aortic isthmus (AoI) narrowing (80% diameter) requires modest adaptations, while severe narrowing (20% diameter) necessitates significant right ventricular (RV) enlargement and ductus arteriosus (DA) dilation to maintain normal pressures, wall shear stresses, and organ perfusion. Validation with ultrasonographic data supports these findings. Notably, the study identifies specific changes in AoI flow patterns, such as delayed retrograde peaks and increased diastolic velocities, which could enhance early CoA diagnosis. However, despite these compensations, overall fetal hemodynamics, including blood pressure and velocities, remain largely unaltered, except for AoI-specific changes.

The specific comments and questions are:

Discussion:

Line 156: It is observed in Fig. S1 A-B that extreme aortic narrowing could lead to overperfusion of the brain without ventricular disproportion. However, this interpretation may be misleading because Fig. S1 only considers two factors: ventricular disproportion and aortic narrowing. In real CoA patients, other compensatory factors (shown in Fig. 1), such as a dilated ductus arteriosus (DA), can alleviate brain overperfusion by reducing the right ventricular (RV) afterload. This should be acknowledged and discussed further.

Line 179: The authors emphasize that velocity patterns in the aortic isthmus show a delayed retrograde flow peak in Fig. 3, Row 3. However, on Fig. 1, there is already a phase difference in retrograde flow peaks among real healthy data.

Line 188 and similar instances: The study simulates "extreme CoA" patients using a mathematical model that incorporates many simplifications, such as assuming circular vessel lumens, constant vessel segment diameters, and no significant remodeling of vascular beds. Prior studies indicate that these assumptions may not hold true for extreme CoA cases, particularly given the dynamic remodeling observed in real patients. These limitations should be highlighted more explicitly.

Line 203: The authors claim that their model is "adequately accurate" but do not thoroughly address the general lack of accurate wall shear stress assessment in lumped parameter models. A more detailed discussion of this limitation is necessary.

Additional Limitation: The study is based on a small cohort, which is a critical limitation, especially considering that CoA is not an exceedingly rare congenital heart defect. A larger dataset would improve the model's robustness and generalizability.

Materials and Methods:

Line 253: The assumption that ventricular depths are comparable requires citation. This is crucial to substantiate the validity of this parameter in the model.

Line 288: The model is based on Pennati et al. (1997) and Garcia-Canadilla's models. However, Pennati’s model is over two decades old, and there have been significant advancements in closed-loop modeling, particularly in heart module development. The authors should justify their choice of an older model over newer approaches.

Line 289: Since the current model is a composite of several previous models, it is essential to describe it in detail in supplementary materials. This would provide transparency and facilitate reproducibility.

Line 297: The non-linear effect described for the AoI compartment should be specified with the corresponding equation to enhance clarity and reproducibility.

Line 310: The middle cerebral artery is described in the text but not depicted in the schematic drawing in Fig. 4. This discrepancy should be addressed for consistency.

Line 342: The authors state that compliance is proportional to the square of the diameter, but Garcia’s earlier work (2014) used a cubic relationship. The justification for this change should be provided.

Line 359: Maximum wall shear stress is calculated using 4μQ/r3, which assumes steady flow conditions. However, fetal circulation is pulsatile, as evidenced by the higher pulsatility indices in CoA patients (Table 1). A wall shear stress equation appropriate for pulsatile conditions should be considered.

Line 375: The validation of the model against real healthy and CoA cases in Fig. 2 should be elaborated upon, including more detailed results and limitations.

Line 383: Details on how ventricular disproportion is simulated are sparse. More information should be provided.

Figures

Figure 1:

This figure validates the mathematical model against real healthy data but lacks validation against real CoA data. Models can often replicate the conditions they are based on (e.g., healthy state), but their validity in other states (e.g., CoA) remains uncertain.

The control group used for validation spans a gestational age range of 6 weeks (31.8 ± 2.89 weeks), which likely includes significant hemodynamic variability due to fetal growth. Normalization or categorization by gestational age is necessary to improve accuracy.

Even for healthy data, the AoI validation, a central focus of this study, does not match real data adequately.

Figure 4:

The meaning of dashed connections should be clarified. Do they represent specific characteristics or assumptions in the model?

The inclusion of specific organs (e.g., kidney, liver, intestines, legs) as separate vascular beds rather than lumping them raises questions. If clinical data were collected for these organs, this should be explicitly stated and validated.

Additional Comments

Data Sharing: A clear data-sharing plan is missing. To enhance reproducibility and foster collaboration, consider providing anonymized datasets or simulation codes through an open-access platform.

In Appendix S1

Line 9 “Moreover, according to Gallivan et al., W (in grams) and GA (in weeks) in healthy human fetuses are related by

log_10⁡W=0.2508+0.1458·GA-0.0016·GA^2.

”

Comments:

This equation appears to have been derived from data presented in previous literature. To enhance clarity and credibility, it would be beneficial to include the original data in a table or diagram, alongside the curve fitting obtained in the current study.

Additionally, it seems there may be a typographical error, as “Gallivan et al.” does not appear to be cited properly.

Line 19: “Due to their unusual diameter growth, peculiar locations, and functions in the fetal cardiovascular system, the scaling factors of the non-linear resistances in the ductus venosus, ductus arteriosus, and foramen ovale were set to -0.88, -2.5, and -0.6, respectively.”

What is the fluid mechanics rationale for setting the scaling factors of the non-linear resistances in the ductus venosus, ductus arteriosus, and foramen ovale to -0.88, -2.5, and -0.6, respectively?

**Have the authors made all data and (if applicable) computational code underlying the findings in their manuscript fully available?**

Reviewer #1: **No: ** No links to relevant code repositories were provided. Please see the feedback to the authors for more details.

Reviewer #2: None

PLOS authors have the option to publish the peer review history of their article (what does this mean? ). If published, this will include your full peer review and any attached files.

**Do you want your identity to be public for this peer review?** For information about this choice, including consent withdrawal, please see our Privacy Policy .

Reviewer #1: No

Reviewer #2: No

**Figure resubmission:**
---

## [Decision Letter · Decision Letter 1]

28 Apr 2025

Dear Ms Villanueva Baxarias,

We are pleased to inform you that your manuscript 'Understanding the hemodynamic changes in fetuses with coarctation of the aorta using a lumped model of fetal circulation' has been provisionally accepted for publication in PLOS Computational Biology.

Best regards,

Andrew D. McCulloch, Ph.D.

Academic Editor

PLOS Computational Biology

Marc Birtwistle

Section Editor

PLOS Computational Biology

Reviewer's Responses to Questions

**Comments to the Authors:**

Reviewer #1: Thank you for your careful revision of the manuscript, which has markedly improved. I recommend that the authors clarify that their model operates at moderate Reynolds numbers and ideally provide a numerical range, since "low Reynolds numbers" (l. 421, p. 21) typically refer to Re much less than 1. Also, I believe you meant to refer to Fig. S3, not Fig. S5, in the Uncertainty Analysis section (Appendix S1, l. 92). I do not have any further comments.

**Have the authors made all data and (if applicable) computational code underlying the findings in their manuscript fully available?**

Reviewer #1: Yes

PLOS authors have the option to publish the peer review history of their article (what does this mean? ). If published, this will include your full peer review and any attached files.

**Do you want your identity to be public for this peer review?** For information about this choice, including consent withdrawal, please see our Privacy Policy .

Reviewer #1: No

---

## [Editor Report · Acceptance letter]

PCOMPBIOL-D-24-01800R1

Understanding the hemodynamic changes in fetuses with coarctation of the aorta using a lumped model of fetal circulation

Dear Dr Villanueva-Baxarias,

I am pleased to inform you that your manuscript has been formally accepted for publication in PLOS Computational Biology. Your manuscript is now with our production department and you will be notified of the publication date in due course.

With kind regards,

Anita Estes
